# Evaluation of First Treatment Timing, Fatal Disease Onset, and Days from First Treatment to Death Associated with Bovine Respiratory Disease in Feedlot Cattle

**DOI:** 10.3390/vetsci10030204

**Published:** 2023-03-08

**Authors:** Kristen J. Smith, Brad J. White, David E. Amrine, Robert L. Larson, Miles E. Theurer, Josh I. Szasz, Tony C. Bryant, Justin W. Waggoner

**Affiliations:** 1Beef Cattle Institute, College of Veterinary Medicine, Kansas State University, Manhattan, KS 66505, USA; 2Adams Land and Cattle Company, Broken Bow, NE 68822, USA; 3Veterinary Research and Consulting Services, LLC, Hays, KS 67601, USA; 4Five Rivers Cattle Feeding, LLC, 4848 Thompson Pkwy #410, Johnstown, CO 80534, USA; 5Southwest Research and Extension, Kansas State University, Garden City, KS 67846, USA

**Keywords:** bovine, bovine respiratory disease, BRD, late-feeding stage, case fatality risk, first treatment success

## Abstract

**Simple Summary:**

Bovine respiratory disease is a frequent and economically important disease in the beef industry. Little research exists describing the expected timing of bovine respiratory disease events and subsequent case outcomes. The objective of this research is to describe temporal patterns of disease occurrence, days to death following treatment, and timing of fatal disease onset. Operational data from 25 participating feed yards were used to create a relevant subset of respiratory disease cases and mortalities for comparison of temporal disease patterns. Results illustrated that the time of arrival to the feed yard influenced the timing of disease onset. The days to death following treatment displayed a right-skewed distribution with cattle arriving in the second quarter dying later after treatment compared to cattle arriving in the third and fourth quarters. Timing of fatal disease onset varied by quarter and displayed a wide distribution of event timing relative to arrival. Understanding typical disease temporal patterns can help cattle health managers appropriately allocate resources for disease control.

**Abstract:**

Bovine respiratory disease (BRD) is a frequent beef cattle syndrome. Improved understanding of the timing of BRD events, including subsequent deleterious outcomes, promotes efficient resource allocation. This study’s objective was to determine differences in timing distributions of initial BRD treatments (Tx1), days to death after initial treatment (DTD), and days after arrival to fatal disease onset (FDO). Individual animal records for the first BRD treatment (*n* = 301,721) or BRD mortality (*n* = 19,332) were received from 25 feed yards. A subset of data (318–363 kg; steers/heifers) was created and Wasserstein distances were used to compare temporal distributions of Tx1, FDO, and DTD across genders (steers/heifers) and the quarter of arrival. Disease frequency varied by quarter with the greatest Wasserstein distances observed between Q2 and Q3 and between Q2 and Q4. Cattle arriving in Q3 and Q4 had earlier Tx1 events than in Q2. Evaluating FDO and DTD revealed the greatest Wasserstein distance between cattle arriving in Q2 and Q4, with cattle arriving in Q2 having later events. Distributions of FDO varied by gender and quarter and typically had wide distributions with the largest 25–75% quartiles ranging from 20 to 80 days (heifers arriving in Q2). The DTD had right-skewed distributions with 25% of cases occurring by days 3–4 post-treatment. Results illustrate temporal disease and outcome patterns are largely right-skewed and may not be well represented by simple arithmetic means. Knowledge of typical temporal patterns allows cattle health managers to focus disease control efforts on the correct groups of cattle at the appropriate time.

## 1. Introduction

Bovine respiratory disease (BRD) is a multifactorial disease with important economic impacts [1,2,3,4]. The detrimental effects of BRD on cattle welfare and performance are accentuated following treatment failure [5]. Previous research has focused on risk factors, weather patterns, diagnostics, disease prediction, and cattle demographics associated with BRD frequency [6,7,8,9,10]. Limited research on the timing of bovine respiratory disease (BRD) onset throughout the feeding phase has been conducted [11,12,13], and little information is available on the timing of disease outcomes relative to treatment. Knowledge of typical temporal disease patterns can help cattle health managers appropriately allocate resources for BRD control.

The timing of BRD initial treatments has previously been studied, and about 75% of BRD cases are reported to occur within the first 40 to 55 days after arrival [13,14,15]. While BRD morbidity risk is greatest early in the feeding phase, the risk decreases toward the end of the feeding period [16]. More recent reports have illustrated BRD also occurs later in the feeding phase [12]. Most previous research focuses on describing the central tendencies (mean/median) for disease timing. Understanding disease timing can be important when instituting BRD prevention and control procedures.

Monitoring treatment failures and understanding potential risk factors can be important knowledge for creating an appropriate therapeutic protocol [17]. Several studies evaluated potential biomarkers and pathogens associated with fatal BRD [2,18,19]; however, few studies have evaluated epidemiologic factors associated with the timing of post-BRD treatment outcomes. Case fatality risk following treatment is often a primary outcome in BRD therapeutic evaluations [20,21], yet, little information is available on the timing from first treatment to death in commercial cattle operations. Research has illustrated the timing of death after arrival may be influenced by the presence of specific pathogens [22], but little information is known about typical temporal patterns of days from treatment to death. Improved knowledge of the timing from first treatment to death could help researchers and cattle health providers design appropriate post-treatment monitoring windows. 

The timing of fatal disease onset is important as fatal cases have the most detrimental outcome. One study reported cattle treated in the first 20 DOF were more likely to be treatment failures than cattle treated after the first 40 DOF; however, this study only evaluated cattle treated up until day 60 and did not specifically evaluate the timing of fatal disease onset [23]. Another study reported that mortality rates have been trending upward from 2001 to 2013, and identified a lower than historical morbidity rate but a higher case fatality rate during this time span [24]. While most BRD morbidity occurs early in the feeding phase, mortality risk has been reported as uniform throughout the remainder of the feeding period [25]. More information is needed about the expected timing of fatal disease onset and subsequent mortality to improve design of BRD therapeutic and preventative measures. 

Common control techniques for BRD include metaphylaxis (antimicrobial administration on arrival) [26,27], vaccinations [28,29], and therapeutic antimicrobials at the time of disease diagnosis. The efficacy of preventative control procedures such as vaccination is dependent on disease and administration timing [30]. Understanding expected BRD morbidity timing and case outcome temporal patterns can lead to an improved ability to evaluate BRD control program effectiveness. The objective of this study was to determine differences in distributions of the timing of initial BRD treatment (Tx1), time of death after initial treatment (DTD), and days to fatal disease onset (FDO) using a subset of data standardized by known risk factors. 

## 2. Materials and Methods

Institutional Animal Care and Use Committee approval was not required as historical operational data were utilized for the analysis and no procedures were performed on cattle specifically for this research.

### 2.1. Data Source

Data were collected from 25 U.S. commercial feed yards under data use and confidentiality agreements. Feed yards were located primarily in the U.S. central high plains region. Event records for 567,989 cattle are included in these data representing a total of 4,381,336 cattle on feed. Data from 11 feed yards were provided for 2015–2019, while data from the other 14 were provided for 2018–2021. Feed yards ranged in one-time capacity from approximately 8000 head to over 100,000 head. 

### 2.2. Data Filtering and Management

Initial data management included the categorization of known BRD risk factors similar to previous work [11,12,23]. Categorized variables included: cohort arrival weight class (226.8–272.2, 272.3–317.4, 317.5–362.9, 363.0–408.2, and 408.3–453.6 kg), the quarter of arrival (Q1, Jan to March; Q2, April to June; Q3, July to Sept; Q4, Oct to Dec), and cohort sex (steer, heifer). 

Data from operational databases were initially filtered to remove any extreme outliers and potential typographical data entry errors. Filtering included only cattle present in cohorts (groups of animals arriving and managed together) with 40 to 400 head at arrival, sex was limited to steer or heifer, and arrival weights from 226.8 to 453.6 kg. Only cohorts with a minimum of 150 days on feed (DOF) were included to avoid cohorts with a very short time at risk for disease. Only treatment or death events occurring before 150 DOF were considered; therefore, the time at risk for disease or death was the same among all cohorts. 

Initial filters limited data to only animals diagnosed with BRD for treatment at least once during the evaluation period. An animal was classified as a BRD treatment if the animal was diagnosed by the feed yard personnel with BRD and received antimicrobial therapy for BRD. First treatment BRD included only the initial BRD therapy and did not include cattle diagnosed with other diagnoses such as acute interstitial pneumonia (AIP). 

Arrival cohort weight has been associated with BRD risk; therefore, a single, common arrival weight category was selected for the analysis (318–363 kg). While this limits the external validity of the findings, this allows evaluation of BRD timing and disease outcomes in a limited subset promoting internal study validity. 

### 2.3. Outcomes of Interest

Three primary outcomes of interest were evaluated in the study: the timing of initial BRD treatment (Tx1), time of death after initial treatment (DTD), and days to fatal disease onset (FDO). The Tx1 represented the number of days from cohort arrival to initial treatment for BRD. Days between initial treatment and death due to BRD were calculated as DTD. The FDO was the date of initial treatment for cases that ultimately had a fatal disease; therefore, FDO is a subset of Tx1. 

### 2.4. Data Analysis

Descriptive graphical analysis and Wasserstein distance (WD) were used to describe, visually evaluate, and calculate the differences in distributions of morbidity and mortality throughout the feeding phase for all three outcomes. Wasserstein distance (WD) is a calculated metric useful for comparing two distributions and the resulting value is equal to the average distance between two corresponding distributions [31]. The unit of analysis is the full distribution and the larger WD, the greater the difference between the two distributions [32,33,34]. Wasserstein distance is equal to 0 when two distributions perfectly overlap and is greater when distributions are farther apart. Comparing distributions can be useful when distributions may be highly skewed and important information can be gained by evaluating the entirety of the study population rather than the central tendency. Wasserstein distances were calculated with the transport package in R Studio. For each subset of data WD, comparisons were calculated for every possible comparison of the quarter of arrival, as well as with each sex compared for every quarter of arrival. 

## 3. Results

### 3.1. Study Population

Raw operational data were provided and filtered to create a dataset for analysis. The data subset used in the analysis included both steers and heifers arriving in cohorts with an average arrival weight between 318 and 363 kg. Results from the filtering process to create the subset for analysis are provided in Figure 1.

The final data for Tx1 analysis comprised cattle meeting the inclusion criteria and treated for BRD at least once (*n* = 102,811). To calculate DTD, a subset consisted of cattle from Tx1 that also died prior to the 150-day inclusion window (*n* = 4663). The FDO subset included all animals in Tx1 that died at any point during the feeding phase (*n* = 4969). The difference in the number of cattle involved in first treatment to death and fatal disease onset is due to the inclusion criteria of cohort DOF being equal to or greater than 150 days, and events (treatment or death) only evaluated for the first 150 DOF.

### 3.2. Days from Arrival to Initial BRD Treatment: Tx1

The Tx1 data (*n* = 102,811) representing the timing to first treatment and distributions of Tx1 ranged by gender and quarter (Table 1). Fewer steers arrived in Q2 compared to the other three quarters.

The timing of Tx1 displayed a right-skewed distribution that varied by quarter and by gender (Figure 2). The temporal patterns varied in shape indicating the concentration of Tx1 varied with a lower concentration early in the feeding period when cattle arrived in Q2. Steers and heifers arriving in Q3 and Q4 displayed a relatively high level of cases early in the feeding phase, then after 50% of cases occurred, the daily risk was relatively low through the 150-day evaluation period.

### 3.3. Days to Death following Initial BRD Treatment: DTD

The DTD following treatment included only cattle with Tx1 that died before the end of the 150-day evaluation period (*n* = 4663). Across all categories, if an animal was going to die from BRD post treatment for BRD, 75% of the time, it occurred within the first 38 days from treatment, except for steers arriving in the Q2, where 75% occurred at DOF 43 (Table 2). Little variability existed among categories for the first 25% of DTD which was less than 3 or 4 days.

Distributions for DTD were visually similar among quarters of arrival and gender (Figure 3). Each distribution contained a high concentration of cases early in the feeding phase, with a tail extending to nearly the end of the evaluation period.

### 3.4. Onset of BRD in Cases with BRD Associated Fatalities (FDO)

Onset of BRD cases resulting in fatal disease (FDO) included cattle in Tx1 that died at any point during the feeding period (*n* = 4969). The range between the 25% and 75% quartiles in FDO varied by gender and quarter of arrival (Table 3). Steers had 75% of their FDO treatments by DOF 66 when they arrived in Q2; comparably, heifers did not reach 75% until DOF 80 for Q2. 

Distributions of FDO varied among genders and quarters (Figure 4). Steers and heifers arriving in Q4 displayed a concentration of FDO cases early in the feeding period and the distribution contrasted with cattle arriving in Q2.

### 3.5. Wasserstein Distances: Tx1, DTD, FDO

Potential differences among distributions for each outcome (Tx1, DTD, and FDO) for each quarter of arrival combination were compared among steers, heifers, and all data combined by evaluating Wasserstein distances (Table 4). The distance between distributions was greatest between Q2 and Q4 in most instances, with relatively small distances present between Q1 and Q2.

A greater difference in Tx1 between steers in Q2 and Q4 was identified compared to any other quarter-comparison with a WD 11.29, but for heifers, the Tx1 difference was greatest between Q2 and Q3 with a WD of 11.42. Both steers and heifers had Tx1 later when they arrived in Q2 compared to the Q1, Q3, or Q4. For DTD, greater WDs were noted in several quarter comparisons relative to Tx1, indicating larger discrepancies between distributions among DTD compared to Tx1. The largest WD in FDO was between Q2 and Q4 for both steers and heifers (17.20; 17.18, respectively). The smallest variation between FDO distributions was seen between Q1 and Q2 for both steers and heifers.

## 4. Discussion

A substantial number of cases were represented in the Tx1 data along with included mortality data to calculate DTD and FDO. While the timing of treatment for BRD relative to feedlot arrival has been described [12,14], little work has evaluated DTD and FDO [22]. Distributions of Tx1 varied by quarter of arrival and DTD illustrated a very concentrated occurrence early with a long-right tail. Understanding FDO is critical as these cases have the most deleterious outcomes and overall disease patterns were visually similar to Tx1 for comparable genders and quarters. Differences in distributions were identified using WD, with greatest differences noted between Q2 and Q4 of arrival for each outcome. Results from this work will help cattle health providers manage expectations and appropriately allocate resources following BRD treatment. 

Visual evaluation of the distribution of Tx1 timing illustrated many cases of BRD occurred early in the feeding period, which agrees with most published research [10]. For both steers and heifers the pattern of disease showed a lower early feeding phase concentration when cattle arrived in Q2. This finding agreed with another study which found an association between timing of disease onset and sex being modified by QOA, with the greatest probability for late-feeding stage morbidity being in heifers that arrived in Q2 [12]. Gender and time of year have both also been associated with Acute Interstitial Pneumonia (AIP), which also tends to occur late in the feeding period [35,36]. The current study only evaluated the timing of BRD morbidity and mortality; however, only retrospective production data were utilized and diagnoses were based on information entered by the operation. Diagnosis of BRD is far from perfect [7,37] and misdiagnosis could occur in this population. Both AIP and BRD would present similar clinical signs; therefore, distinguishing ante-mortem may be challenging. Post-mortem examinations may be useful to delineate the two syndromes which would provide further distinction and information relative to timing of disease. 

Although differences in the timing of Tx1 distributions were evident, the distributions of DTD were visually very similar among quarters and genders. Many of the cases occurred very soon after Tx1 with 25% of deaths occurring within 3 or 4 days of treatment, and these results quantify conceptual expectations of timing of treatment failure [5]. Similarity of DTD patterns among quarters of arrival and gender is interesting as the disease has been reported to occur later in the feeding period for heifers and Q2 arrivals [12]. The current research illustrated that the difference in mortality timing is likely more related to Tx1 than DTD based on the distribution of these two variables. Potential hypotheses for the early occurrence of death following treatment is related to a failure to diagnose BRD early in the disease process or severe disease non-responsive to therapy [17]. Describing population DTD allows animal health managers to create interventions if the distribution differs from expected in this type of cattle. This information is valuable for animal health providers when considering the overall treatment protocol and timing of potential retreatment. Retreatment events were not considered in this study, but after the initial phase that accounts for 50% of deaths, the daily risk for death was very small for a large portion of the evaluation period. 

An average, FDO has been reported at 32.65 days with differences in FDO with presence of viral pathogens associated with earlier FDO [22]. This reported FDO average is similar to medians reported in the current study; however, the previous work did not account for differences among genders, arrival cohort weight, or arrival time of year. The wide ranges in FDO in this study could be related to differences in pathogens or disease processes, and improved understanding of these factors could lead to improved treatment and control. Identifying FDO patterns outside of expectations may lead to earlier interventions by cattle health providers.

Comparison of distributions using WD revealed the greatest differences between cattle arriving in Q2 and Q4 for Tx1 and FDO. More animals arrived in Q1, Q3, and Q4 than in Q2, which is typical for the industry due to the seasonal timing of calving with most U.S. calves born in the spring. Onset of disease (fatal and non-fatal) displayed a much different pattern in Q2, with no distinct peak or major disease period. This finding is interesting and may be due to differences in cattle, pathogens, or other seasonal factors. Previous work has illustrated that the risk of BRD is higher in cattle arriving in Q3/Q4 [1,38]; however, a difference in the timing of the disease events has not been well-documented. One potential hypothesis is the difference in timing is related to the overall risk level of cattle, as one report indicates that lower risk cattle may contract disease later in the feeding period [11]. Cattle arriving in Q2 may be at lower risk due to decreased density of cattle arriving at that time or a difference in age post-weaning relative to cattle arriving in Q3 and Q4. More research should be performed to further elucidate potential risk factors and causes for cattle in this arrival time frame. A better understanding of disease patterns in this timeframe may lead to better BRD treatment and control options. 

One limitation of this work is the limitation of evaluated data to one weight class of cattle. Cattle in different weight classes face differing levels of BRD risk. Limiting to a single weight class improves internal validity but limits the external validity of the study to only this specific population. Additionally, while multiple feed yards are represented, results do not necessarily represent the entirety of the U.S. cattle feeding industry which limits the appropriateness of wider extrapolation from these data. 

## 5. Conclusions

This research evaluated outcomes from BRD cases in commercial U.S. cattle feeding operations to determine temporal distributions of Tx1, DTD, and FDO for steers and heifers weighing 318–364 kg for each quarter of arrival. These distributions varied by gender and quarter of arrival for each outcome. Temporal patterns were right-skewed and graphic visualization of each outcome provides information for animal health decision makers to modify plans for identifying new BRD cases, as well as appropriate case outcome evaluation periods. Comparison of WD among outcomes illustrated that expectations should change among time of year based on the outcome of interest. The outcomes of this research provide expectations for the timing of adverse events following BRD treatments, which can be helpful to cattle health managers.

## Figures and Tables

**Figure 1 vetsci-10-00204-f001:**
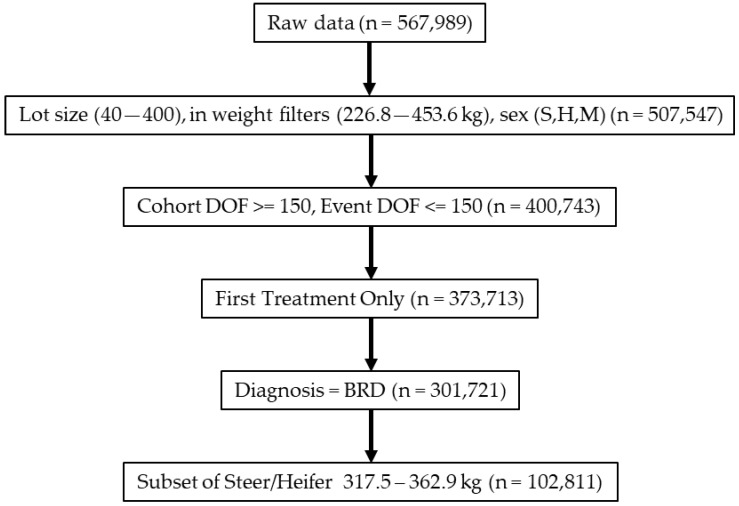
Flowchart of data filtering process for the number of individual animals filtered out to create the working dataset for the morbidity timing analysis.

**Figure 2 vetsci-10-00204-f002:**
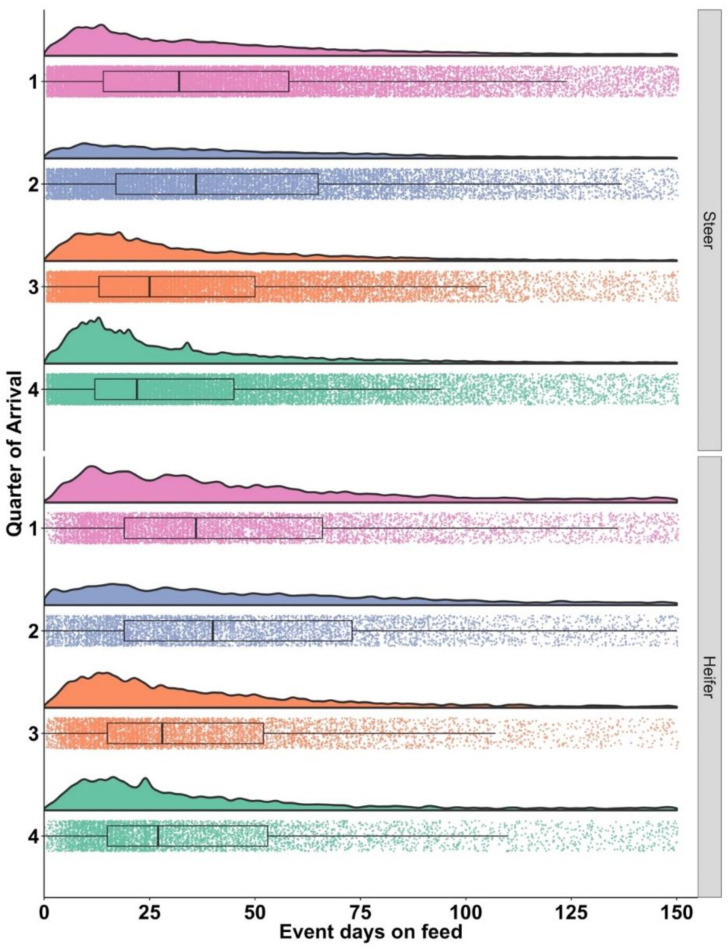
Days on feed from arrival to first treatment for bovine respiratory disease (Tx1), by quarter of arrival on the left vertical axis and sex faceted on the right vertical axis. Each arrival quarter is represented by a different color (light red = Q1, blue = Q2, orange = Q3 and green = Q4). The vertical axis is scaled to account for the difference in number of observations in each category. Each dot represents one observation. Box and whisker plot box represent the upper and lower quartile with the heavy line inside the box being the median.

**Figure 3 vetsci-10-00204-f003:**
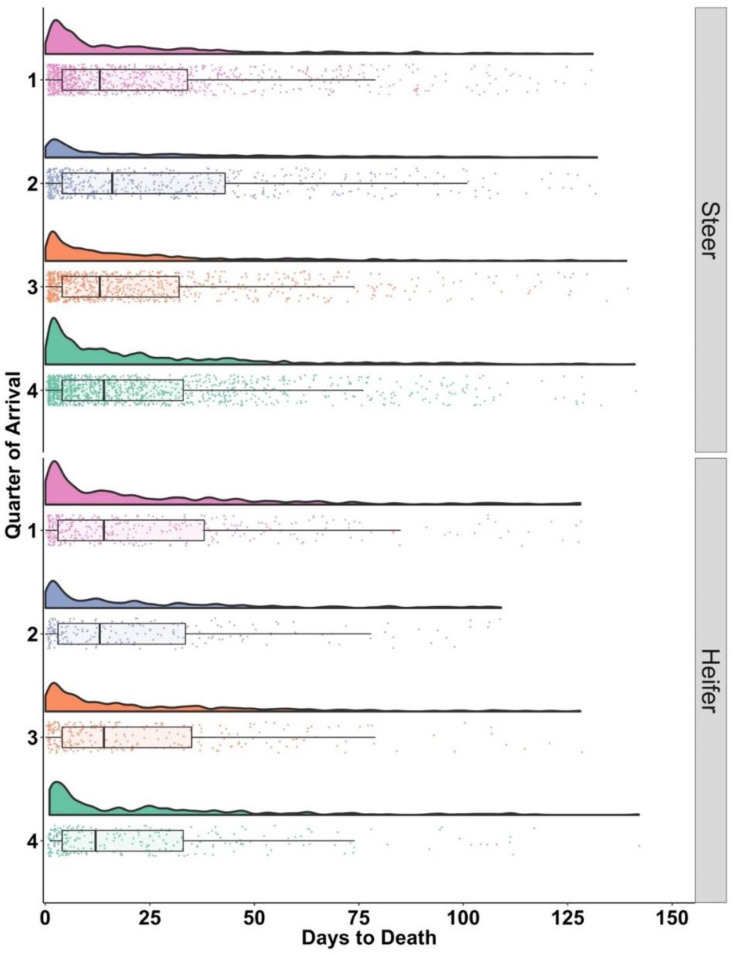
Days from first treatment to death, by quarter of arrival on the left vertical axis and sex faceted on the right vertical axis. Each arrival quarter is represented by a different color (light red = Q1, blue = Q2, orange = Q3 and green = Q4). Box and whisker plot box represent the upper and lower quartile with the heavy line inside the box being the median. Day 0 here refers to the day of first treatment for bovine respiratory disease.

**Figure 4 vetsci-10-00204-f004:**
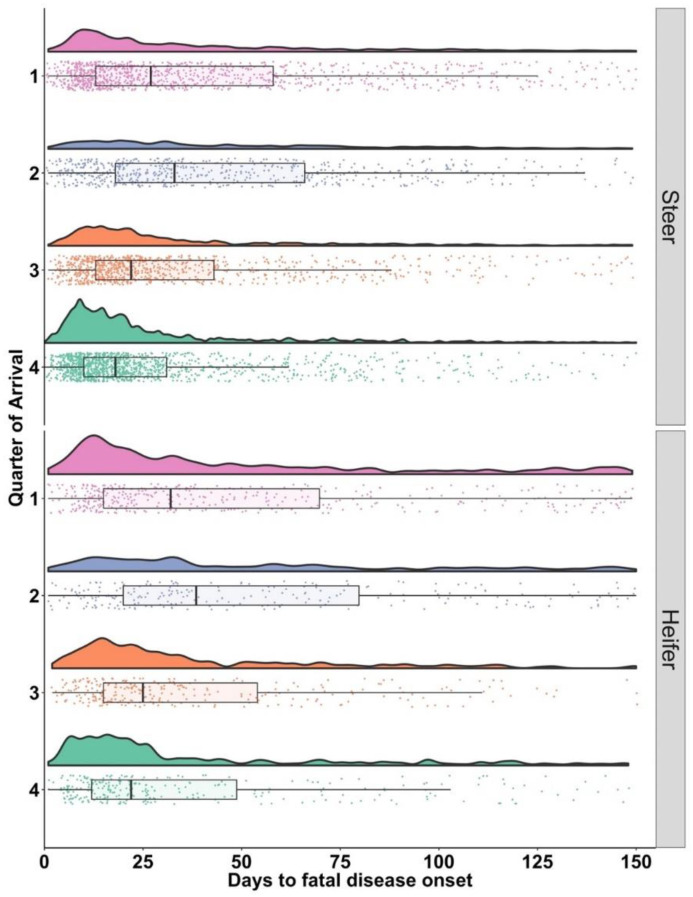
Days on feed from arrival to first treatment for bovine respiratory disease (BRD) for animals that eventually died from BRD (FDO), by quarter of arrival on the left vertical axis and sex faceted on the right vertical axis. Each arrival quarter is represented by a different color (light red = Q1, blue = Q2, orange = Q3 and green = Q4). Box and whisker plot box represent the upper and lower quartile with the heavy line inside the box being the median.

**Table 1 vetsci-10-00204-t001:** Days on feed quartiles for bovine respiratory disease first treatment by gender and quarter of arrival in a subset of cattle with an average arrival weight between 317.5 kg and 362.9 kg (*n* = 102,811).

Characteristic		Tx1 (DOF)	
Quarter of Arrival		Quartiles	
Steers	*n*	25%	75%
1	22,678	14	58
2	13,946	17	65
3	19,342	13	50
4	23,567	12	45
Heifers			
1	7223	19	66
2	5224	19	73
3	5627	15	52
4	5204	15	53

**Table 2 vetsci-10-00204-t002:** Days to death following bovine respiratory disease treatment (DTD) quartiles by gender and quarter of arrival in a subset of cattle with an average arrival weight between 317.5 kg and 362.9 kg (*n* = 4663).

Characteristic		DTD (DOF)	
Quarter of Arrival		Quartiles	
Steers	*n*	25%	75%
1	927	4	34
2	557	4	43
3	806	4	32
4	1257	4	33
Heifers			
1	307	3	38
2	215	3	34
3	261	4	35
4	270	4	33

**Table 3 vetsci-10-00204-t003:** Days on feed at initial treatment for bovine respiratory disease in fatal cases (FDO) quartiles by gender and quarter of arrival in a subset of cattle with an average arrival weight between 317.5 kg and 362.9 kg (*n* = 4969).

Characteristic		FDO (DOF)	
Quarter of Arrival		Quartiles	
Steers	*n*	25%	75%
1	1009	13	58
2	605	18	66
3	847	13	43
4	1309	10	31
Heifers			
1	398	15	70
2	232	20	80
3	275	15	54
4	284	12	49

**Table 4 vetsci-10-00204-t004:** Wasserstein Distance used to compare distances between distributions for days on feed by sex and quarter of arrival for first treatments for bovine respiratory disease (Tx1), days to death after initial treatment (DTD), and days after arrival to onset of bovine respiratory disease in fatal cases (FDO). Greater differences between quarters of comparison are seen by greater values of Wasserstein Distance.

Quarter of Comparison	Wasserstein Distance		
	First Treatment (Tx1)	Days to Death (DTD)	Days to Fatal Disease Onset (FDO)
Steers			
1–2	3.63	6.96	4.99
1–3	6.06	6.41	6.46
1–4	7.78	12.00	12.88
2–3	9.56	13.21	10.73
2–4	11.29	18.90	17.20
3–4	9.56	5.70	6.56
Heifers			
1–2	3.10	6.38	6.90
1–3	8.72	9.41	9.30
1–4	7.39	12.05	10.40
2–3	11.42	14.93	15.94
2–4	10.13	17.74	17.18
3–4	1.78	4.53	4.10
Combined			
1–2	3.57	6.67	5.29
1–3	6.75	7.52	7.40
1–4	8.05	13.00	13.19
2–3	10.21	13.88	12.30
2–4	11.51	19.45	18.15
3–4	2.15	5.58	5.91

## Data Availability

Data utilized for this research were from cooperating entities and are not available publicly due to confidentiality and anonymity agreements.

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
