# Peer review of "Evaluation of First Treatment Timing, Fatal Disease Onset, and Days from First Treatment to Death Associated with Bovine Respiratory Disease in Feedlot Cattle"

_vetsci, 2023, doi:10.3390/vetsci10030204_

Round 1

Reviewer 1 Report

The submission "Evaluation of first treatment timing, fatal disease onset, and days from first treatment to death associated with bovine respiratory disease in feedlot cattle" provides some very important issues about one of the most important cattle disease. Overall, the introduction describes all major aspects to be discussed in the end. The material and methods is well written and can be reproduce. The results show each data ccordind to what have be done, Tables and Figures are in agreement to authors instructions. The discussion provides a precise comparison among all results worldwide. 

Reviewer 2 Report

This manuscript describes the temporal distributions of bovine respiratory disease (BRD) presentation and outcomes including days at initial BRD treatment, days to death following initial treatment and onset of fatal BRD disease in feedlot beef cattle with an arrival weight of 318-363 kg. Quarter of arrival and sex were associated with differences in the temporal distributions of BRD initial treatment, days to death following treatment, and fatal disease onset. Two important limitations of this study are that the analysis was only performed in heavier cattle (318-363 kg) and that the definition of a BRD case/treatment was based on each feedlot's information. Although the authors recognize the potential effect of these factors in the external validity of the study, it is important to note that the bulk of economic losses due to BRD occur in high-risk lighter cattle and that inaccurate definition of BRD and treatment could have confounded the results of this study. In the opinion of this reviewer, this manuscript advances the knowledge on temporal BRD distribution in the feedlot and contributes to the establishment of appropriate management interventions when known temporal BRD distribution altered.